# OCT-Angiography Findings in Children with Anisometropic Amblyopia

**DOI:** 10.3390/children10091519

**Published:** 2023-09-07

**Authors:** David Pekica, Nina Košič Knez, Barbara Razboršek, Dušica Pahor

**Affiliations:** 1Faculty of Medicine, University of Maribor, Taborska ulica 8, 2000 Maribor, Slovenia; david.pekica@student.um.si (D.P.); knez.nina@gmail.com (N.K.K.); barbara.crnjac@student.um.si (B.R.); 2Department of Ophthalmology, University Medical Centre Maribor, Ljubljanska ulica 5, 2000 Maribor, Slovenia

**Keywords:** OCT angiography, macula, anisometropia, amblyopia, children

## Abstract

Purpose: The purpose of this prospective study is to show findings of OCT angiography in children with anisometropic amblyopia with a statistically significant difference, regardless of the type of refractive disorder, between the amblyopic and the fellow eye. This research aimed to establish whether there is a difference in vascular density [VD] and size of the foveal avascular zone [FAZ] in the superficial capillary plexus [SCP]. Methods: All children between 9 and 18 years of age who were treated at the Outpatient Clinic for Orthoptics and Pleoptics of the Department of Ophthalmology, University Medical Centre Maribor from January 2020 to December 2022 due to unilateral anisometric amblyopia were enrolled in our study. Fourteen children met the criteria. Cirrus 5000 high-resolution OCT with AngioPlex OCT angiography was used to analyze the size of the FAZ and VD in the SCP and TCS. The paired *t*-test or Wilcox signed-rank test [*p* < 0.05] was used for statistical analysis of each parameter between the amblyopic and fellow eye. Results: Mean child age was 13 years ± 2.9 and ranged from 9 to 18 years. Most of the children [85.7%] were boys. The VD of the SCP did not show statistically significant differences between the visually impaired and control eyes [*p* = 0.328]. The comparison of the FAZ area between the two eyes was also not statistically significant [*p* < 0.808]. There was also no statistically significant difference in central macular thickness [TCS] [*p* < 0.291]. Conclusions: Our research results show no statistically significant differences in the VD and the FAZ of the SCP, and in the TCS between the amblyopic and fellow eye in children with unilateral anisometropic amblyopia. Our research did not confirm the results of certain previous studies in which a lower density of the capillary network was present in the visually impaired eye. Further studies with more children are necessary to confirm our results.

## 1. Introduction

Amblyopia is the most common monocular and binocular vision deficit in children [1]. The estimated prevalence of amblyopia is from 0.5 to 3.7% [1,2,3,4]. A previous meta-analyses study consisting of 73 studies published in 2018 showed a prevalence of amblyopia in 1.75% of children worldwide, varying from 0.51% in Africa to 3.67% in Europe [2]. A meta-analysis published in 2019 consisting of 60 studies reported a 1.44% prevalence of amblyopia, ranging from 0.72% in Africa to 2.90% in Europe [3].

A meta-analysis published in 2022 consisting of 97 studies for the global prevalence of amblyopia in children showed that the estimated prevalence was 1.36%. When considering each continent, the prevalence of amblyopia was 2.66% in total, 1.95% in North America, 1.86% in Oceania, 1.16% in Asia, 0.46% in South America, 0.38% in Africa, and 0.76% in mixed countries [4]. 

Amblyopia is defined as a permanent visual reduction due to the failure of cortical visual development due to abnormal visual experiences in childhood. Amblyopia is a neurodevelopmental disorder with no ocular pathology. It is diagnosed as reduced vision in one or both eyes without any abnormality of the eye with an interocular visual difference of two lines or more. The cause for amblyopia is stimulus deprivation as seen in congenital cataracts and congenital ptosis. Strabismus and anisometropic refractive errors are the most common causes of amblyopia in children [5,6].

Amblyopia is treated by eliminating the underlying cause of visual deprivation and correcting the refractive error with glasses or contact lenses. The goal is to achieve the same visual acuity in both eyes. The prognosis of therapy is better in young children. The outcome of treatment is influenced by various factors such as initial visual acuity, types of low vision, previous therapy, eye comorbidities, and compliance to treatment [7,8]. 

Correction of the refractive error is the most important step in the treatment of amblyopia. Primary optical correction without other measures is effective in almost one third of cases. Occlusion is an important method in the treatment of amblyopia. The eye with better visual acuity is covered for several hours for better vision development of the amblyopic eye depending on the degree of amblyopia. Holmes et al. found out that the effect of 6 h and all-day coverage in severe amblyopia was similar [9]. The treatment method also consists of treating the better eye using 1% Atropine drops. With occlusion, we want to achieve greater use of the amblyopic eye. 

In recent years, changes in the primary and visual cerebral cortex were found in children with refractive amblyopia and in the retina [10,11,12,13,14].

Several previous studies revealed alterations in the visual pathways and visual cortex in amblyopic eyes, but the findings of retinal changes in amblyopic eyes are not well-established and remain controversial [15].

Anisometropia is defined as a condition of asymmetric refraction between the two eyes. This condition is defined by a difference of one or more diopters in spherical equivalent. To avoid the development of amblyopia in children, it is very important to diagnose this condition as soon as possible. The prevalence of anisometropia is more pronounced in newborns and can be reduced in the first years of life. The term anisometropic amblyopia is reserved for amblyopia in the eyes with severe refractive error compared to other eyes without other pathologic conditions. Anisometropic amblyopia can be found in individuals who have experienced defocused retinal images due to refractive errors [16,17].

Aniseikonia occurs as a result of the anisometropic refractive error. It is the difference in perceived image size between the right and left eye. In the context of anisometropia, aniseikonia can be caused by anatomical differences in axial length, differences in photoreceptor spacing between the eyes, or cortical adaptations; optically, it can also be caused by the correction of the eye or contact lens prescribed for anisometropia [18]. 

With the advances in optical coherence tomography [OCT] and OCT angiography [OCTA] as new non-invasive diagnostic procedures, further investigations of retinal changes were possible without exposing the patients to any risk and can also be performed for amblyopic eyes in children [19,20,21].

OCTA is based on optical coherence tomography (OCT) which works by passing a beam of laser-generated light, with wavelength in the infrared range, through a tissue and analyzes its reflected fraction. Since the speed of light is too great to measure the difference, the phenomenon of interference is used. The light beam is split into two parts: the first part is directed into the eye to the retina and the second part is directed to the reference mirror. The first part that was directed into the eye is reflected back into the device with changed amplitude. The second beam remains unchanged and is reflected from the reference mirror. When the beams meet, they form an interference signal that is intercepted and measured using the photodetector. OCT measures signals and takes pictures of retinal structures. The OCTA scan is based on the differences between OCT B-scans related to the movement of erythrocytes in the vascular system [22,23].

OCTA, without using invasive dye injection, is a new safety method for retinal vascular investigations. With this method, the foveal avascular zone, vessel density, as well as the retinal perfusion can be examined with no risk for complications associated with the procedure. From this point of view, it can also be used in children [22]. 

With the introduction of OCTA in retinal investigations, many researchers evaluated macular vessel density of superficial and deep capillary plexus also in amblyopic eyes, but only a few reports were published about retinal vascular changes in anisometropic amblyopic eyes in children. The results of these studies are different in their findings, from no changes in vascular density to the reduction of vascular density [23].

The purpose of this current prospective study is to evaluate the features of OCTA in children with unilateral anisometric amblyopia, regardless of the type of refractive disorder, with a statistic significant difference regarding the sphere and spheric equivalent between the amblyopic and the fellow control eye. This research aimed to establish whether there is a difference in vascular density [VD] and size of the foveal avascular zone [FAZ] in the superficial capillary plexus [SCP] and in central retinal thickness between amblyopic and fellow eye [TCS].

## 2. Materials and Methods

All children between 9 and 18 years of age who were treated at the Outpatient Clinic for Orthoptics and Pleoptics of the Department of Ophthalmology, University Medical Centre Maribor from January 2020 to December 2022 due to unilateral anisometric amblyopia were enrolled in our study.

Amblyopia was defined as a condition in which the BCVA was less than 0.8 in the amblyopic eye due to anisometropia, including anisometropia combined with strabismus.

Children aged 9–18 years with unilateral anisometropic amblyopia with the intraocular difference in refractive error of 4.0 diopters between amblyopic and fellow eye or more were included in this study.

The exclusion criteria included any previous history of ocular or head trauma, intraocular surgery, or laser therapy. Children with systemic, ocular, and neurological diseases that may affect visual acuity or cause changes in fundus microcirculation were also excluded from this study, as well as children with binocular amblyopia and nystagmus. The children unable to cooperate during the examination were also excluded.

The presence or absence of a history of amblyopia treatment at the OCTA examination was not considered in the data analysis.

All children who fulfilled the inclusion criteria underwent comprehensive ophthalmological examination including eye position; eye movements; best corrected visual acuity [BCVA] with a Snellen chart; cycloplegic refraction determination with sciascopy; slit-lamp biomicroscopy; fundoscopy in mydriasis; central fixation test; and orthoptic evaluations, including synoptophore examination for squint angle measurements and biometry—axial length [AL] and anterior chamber depth [ACD]. Demographic data including age and sex were recording. 

At the end, a total of 28 eyes, 14 anisometropic amblyopic eyes, and 14 fellow non-amblyopic eyes of 14 children aged from 9 to 18 years with a diagnosis of amblyopia due to anisometropia of 4 diopters or more who met the inclusion criteria were enrolled in this study. Fellow non-amblyopic eye was used as control eye.

All ocular measurements were performed by an experienced examiner. All children received a topical mydriatic agent. Fundus photography and fundus autofluorescence were used to exclude other ocular diseases. 

OCTA images were evaluated by experts for OCT and OCTA in amblyopic and non-amblyopic eyes. All images with poor quality were excluded from the analyses.

OCTA images of the parafoveal areas of the observed eyes were acquired with a Zeiss Cirrus 5000 high-resolution OCT angiography with built-in AngioPlex v. 10 software with a pattern of 6 × 6 mm (Figure 1). 

OCT parameters were as follows: 825 nm wavelength light source, in-tissue A-scan depth of 2.0 mm, resolution of 15 × 5 µm, and a scan speed of 68 k A-scans/second. From the OCTA scans, we obtained data on the FAZ size and VD in the SCP. We used the data for the SCP because the built-in AngioPlex software can identify and measure the blood flow density parameters of the superficial retina in the macular area. 

The SCP was defined with an inner boundary of internal limiting membrane and an outer boundary at the inner plexiform layer. This measurement was performed automatically using built-in AngioPlex v. 10 software. In this way, we were able to ensure objective data. The data from internal limiting membrane to the inner plexiform layer/inner nuclear layer interface for the SCP was independent from the investigator. We did not use the data from the DCP (from the inner plexiform layer/inner nuclear layer interface to the outer plexiform layer/outer nuclear layer interface) because they were not performed automatically and were not included in the AngioPlex software. In our study we wanted to use parameters that can be exactly measured without any personal influence and ones that could detect even the smallest differences.

The FAZ area was defined as the vascular area in the center of the fovea. VD, based on settings, was defined as the total length of perfused vasculature per unit area in a region of measurement [mm/mm^2^]. To determine central macula thickness [TCS], the Macula Cube 200 × 200 program was used. 

For biometry, axial length [AL], and anterior chamber depth [ACD], noncontact partial coherence interferometry evaluation [IOL Master 700, Carl Zeiss Meditec, AG, Jena, Germany] was used.

Statistical analyses were performed using the program JASP [Version 0.16.4]. Decimal BCVA was converted to logMAR for analysis. All data between amblyopic and fellow eyes were compared using the paired *t*-test or Wilcoxon signed-rank test. The test was chosen based on variable normality, which was determined using the Shapiro–Wilk test for all variables. The data were presented as the means and ± standard deviations, or medians [minimum to maximum value]. For all analyses, *p*-values less than 0.05 were considered statistically significant.

This study’s protocol adhered to the tenets of the Declaration of Helsinki and was approved by the Institutional Medical Ethics Commission at the University Clinical Centre Maribor [registration number: UKC-MB-KME-72/22]. They assessed that this research was ethically acceptable and granted approval to conduct this study. Written informed consent was obtained from the parent/legal guardian for participating children.

## 3. Results

Twenty-eight eyes of fourteen children with unilateral anisometropic amblyopia were enrolled in this study, fourteen eyes with anisometropic amblyopia, and fourteen fellow non-amblyopic eyes. The fellow non-amblyopic eyes were used as a control group in our study. 

Strabismus was observed in three of the fourteen children [21.4%], two had anisometropia combined with manifest esotropia [14.3%], and anisometropia combined with manifest exotropia was present in one child [7.1%]. Among the fourteen children, twelve were boys [85.7%] and two [14.3%] were girls. 

The mean age of children was 13.1 ± 2.9 years, ranging from 9 to 18 years.

The clinical parameters are summarized in Table 1. 

The average BCVA expressed in logMAR was significantly lower in amblyopic eyes [0.58 ± 0.50] compared to fellow eyes [0.02 ± 0.06, *p* = 0.002].

The average refraction expressed in sphere was 8.03 ± 4.02 diopters in amblyopic eyes and 2.10 ± 1.52 diopters in fellow eyes. The difference between amblyopy in fellow eyes was not significant [*p* < 0.001].

The average refraction expressed in spherical equivalence was 8.19 ± 4.57 diopters in amblyopic eyes and 2.18 ± 1.55 in fellow eyes. The difference between amblyopic and fellow eyes was not significant [*p* < 0.001] [Table 2].

The average axial length was 24.00 ± 3.85 in amblyopic eyes and 23.47 ± 1.08 in fellow eyes. The difference was not statistically significant [*p* = 0.715].

The average vascular density in superficial capillary plexus [VD SCP] was 18.32 ± 1.22 in amblyopic eyes with a median value of 18.80 [15.20 to 19.30] and 18.90 ± 0.62 in fellow eyes with a median value of 19.00 [17.80 to 19.70] in fellow eyes. The difference between amblyopic and fellow eyes was not significant [*p* = 0.328].

The average size of the superficial foveal avascular zone [FAZ SCP] measurements were 0.227 ± 0.07 mm^2^ in amblyopic eyes with a median value of 0.21 [ranging from 0.13 to 0.35] and 0.232 ± 0.10 mm^2^ in fellow eyes with a median value 0.22 [ranging from 0.09 to 0.45]. The difference between amblyopic and fellow eyes was not significant [*p* = 0.808].

The average central macular full thickness [TCS] was 261.50 ± 36.63 µm in the amblyopic eye with a median value of 259.50 [197.00 to 347.00] and 254.50 ± 23.14 µm in fellow eyes with a median value of 258.00 [215.00 to 288.00]. The difference between amblyopic and fellow eyes was also not significant [*p* = 0.281].

The results are presented in Table 3.

With the exception of outer temporal macular part [OT, *p* = 0.010], no statistic significant differences were found in macular thickness. In amblyopic eyes average value was 281.00 ± 34.23 and 258.29 ± 18.07 in fellow eyes. The results are presented in Table 4.

Regarding the VD in SCP no statistic significant difference was found in different macular locations between amblyopic and fellow eyes [Table 4].

## 4. Discussion

Previous studies revealed different findings in amblyopic eyes on retinal vascular density, with non- to significant changes of lower vascular density of the SCP to larger FAZ area. The aim of our study was to contribute to this unclearly established field of OCTA findings in anisometropic amblyopic eyes. For this proposal, we included the eyes with severe anisometropia [four diopters or more] for better evaluation of OCTA findings between anisometropic amblyopic and non-amblyopic fellow eye. To our knowledge, our study is the first study to compare OCTA findings in children with severe anisometropia. 

The introduction of a relatively new non-invasive OCTA method for retinal evaluation including vascular structure of the retina and choroid gives us a new possibility to perform this method for retinal evaluation also in children.

In recent years, OCTA is more and more used in the evaluation of retina vascular circumstances. OCTA has the ability to evaluate all four main retinal vascular plexus—peripapillary, superficial [SCP], intermediate [ICP], and deep capillary plexus [DCP]—as well as the foveal avascular zone [FAZ]. The FAZ is characterized by a lack of blood vessels. It can be enlarged in many pathologic conditions and is very sensitive to hypoxia [24,25].

Nowadays, different OCTA systems are used with different software and algorithms, which can also be the reason for different results of retinal vascularization [20,25,26,27,28,29,30].

Our findings suggest that amblyopia does not affect vascular density in the superficial capillary plexus, the size of the superficial foveal avascular zone, and the central macular full thickness. The results of other studies using OCTA were conflicting, but more studies suggest that effects in vascular structures are present [13,31,32,33,34,35,36,37].

Different results regarding OCTA findings in amblyopic eyes in children may be due to the fact that the same measurement methods were not used. There were already significant differences in results when 6 × 6 or 3 × 3 volume scans were used [31].

One important limitation factor in our study was the small number of patients. In the majority of previous studies for OCTA findings in amblyopic eye the number of children was also small. However, it must be emphasized that the limited sample size is one of the important limitations of our study. It is a single-centered study, which is also a limitation. We used non-amblyopic fellow eyes as a control; this comparison could be an additional limitation factor.

The selection of our patients was well defined regarding the amount of refractive error between amblyopic and non-amblyopic eye. Difference between amblyopic eyes and non-amblyopic eyes for four or more diopters was important inclusion criteria. The other eye in all participants was non-amblyopic with visual acuity at less than 0.8, or more with correction.

As shown in the meta-analysis by Gao et al., it is not the same when amblyopic eyes are compared to fellow eyes and when they are compared to the eyes of the control group [33]. Vascular density in the SCP and DCP was lower in amblyopic eyes compared with the healthy control eyes, but the difference to the fellow eyes was not significant [33]. They concluded, after the analysis of previous studies, that reduced vascularization in amblyopic eye can also be observed in the fellow non-amblyopic eye. Unilateral amblyopia can also affect the healthy fellow eye. From that point of view, comparison between amblyopic and fellow eye and between amblyopic and control eye is not equivalent. There was also a difference in using different measurements methods such are 6 × 6 or 3 × 3 volume scans. 

Visual development is completed by age six to age seven. Abnormal stimulation of the amblyopic eye could influence lower oxygen supply and decreased vascularization. The consequences can include a thicker fovea during reduced apoptosis of the ganglion cells [16]. The limitation of this meta-analysis was not including non-English studies. In the majority of studies, the number of included children was small. The severity of amblyopia and the types of amblyopia were not considered in the final conclusion. The meta-analysis by Yang et al. revealed similar data. Vascular density was reduced in amblyopic eyes compared to controls [31]. All meta-analyses had several limitations. The most important being the small number of studies and the quality of included studies.

To determine all parameters and reduce the factor of human error, we used built-in software, which is also not the same in all studies. In some studies, the FAZ was measured manually by examiners [36]. Different measurement methods lead to inconsistent results.

Kurumoğlu et al. investigated macular and disk vessel density parameters in healthy children aged 7–18 using OCTA and contributed to normal reference ranges for healthy children. The results of this study could be useful for further studies of pathological conditions in children [27]. 

Previous studies revealed that amblyopia has no effect on retinal thickness. OCTA revealed no significant vascular damage in amblyopic eyes [38,39]. Demirayak et al. found no difference between amblyopic eyes, controls, and fellow eyes of patients [40].

On the contrary, other researchers found vascularity changes in amblyopic eyes. 

Araki et al. found that, although the amblyopic eyes had a smaller FAZ area of the SCP, there was no significant difference in the macular vessel density between the amblyopic and fellow eyes [36]. 

Assessing the foveal avascular zone and vessel density of the retina, Ye et al. found that the foveal avascular zone was larger in amblyopic eyes with lower vascular density in superficial and deep capillary plexus layers in amblyopic eyes, but the difference was not statistically significant [34].

Yilmaz et al. concluded that the underuse of the amblyopic eyes can induce retinal or choroid microvasculature changes [41].

In the study by Sobral et al. on 26 children with amblyopia, a statistically significant decrease in macular vascular density [*p* = 0.0171] of the superficial capillary plexus [SCP] was observed [42].

Feng at al. published the results of a meta-analysis regarding microvascular features in patients with amblyopia based on OCTA, and found out that in the majority of studies on retinal vessel density were lower in amblyopic eyes and also in fellow eyes compared to normal eyes, and concluded that differences are easily found using 6 × 6 mm scans [21].

Unilateral amblyopia can influence contralateral fellow eye and for this reason the contralateral eye is probably not suitable as a control eye [42]. 

For the evaluation of microvascular features in amblyopic eyes, further studies are necessary with an exact definition of the inclusion criteria for investigated eyes, including refractive error, the extent of anisometropia, ethnic heterogeneity, and the age of the children. Younger children should also be included in the studies and in the investigation to contribute to a better understanding of microvascular changes in amblyopic eyes.

Our results revealed no changes in retinal structure in OCTA; therefore, we can conclude that anisometropic amblyopia is not associated with structural changes of the retina, especially with vascular density changes in the SCP, and that the morphological changes do not predispose or contribute to the development of amblyopia. 

We are aware that the most important limitation of our study was the small number of included children. As we already mentioned in the discussion, we excluded all anisometropia eyes with refractive error less than four diopters for better detection of the difference in OCT-angiography changes between amblyopic and fellow eyes.

In the future, we want to perform another study with children with smaller anisometropia ranging from one to two diopters and recruit a higher number of children. Therefore, we can also expect different results as a consequence. 

Further multicenter studies with a larger number of children with unilateral anisometropic amblyopia without strabismus, same inclusion and exclusion criteria, same diagnostic procedure with comparison between three groups—amblyopic, fellow non-amblyopic, and control eye—will give us a real result and confirm or reject the hypothesis of morphological changes in amblyopic eyes evaluated using OCTA.

## 5. Conclusions

Our research results show that there were no statistically significant differences in the density of the macular superficial capillary plexus with the exception of the outer temporal part between amblyopic and non-amblyopic fellow eye. We found no difference in the size of the avascular cone superficial foveal avascular zone and the central macular thickness between the amblyopic and fellow eye in children with unilateral anisometropic amblyopia. Our research did not confirm the results of some previous studies that indicated a lower density of the capillary network being present in the visually impaired eye. Further studies with a higher number of children using the same diagnostic procedures and well-defined inclusion criteria are necessary to confirm our results.

## Figures and Tables

**Figure 1 children-10-01519-f001:**
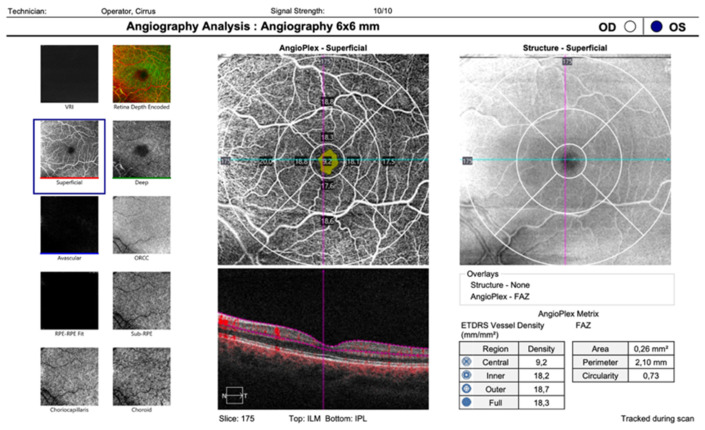
Cirrus OCTA analysis screening—OCTA of an amblyopic eye. On the left side of the screen, a legend of all instances of macular retinal vascular analysis are displayed; illustrated at the bottom of the image is the marked chosen program for superficial analysis. Central part: result from 6 × 6 mm macular scans of vessel density in the superficial retinal capillary plexus of an amblyopic eye (in the central part of the screen) and corresponding OCT image.

**Table 1 children-10-01519-t001:** Clinical data of the participants.

	Amblyopic Eyes No = 14	Fellow Eyes No = 14	*p*-Value
BCVA [logMAR]	0.58 ± 0.50	0.02 ± 0.06	0.002 ^b^
Sphere [diopters]	8.03 ± 4.02	2.10 ± 1.52	<0.001 ^a^
Spheric equivalent refraction [diopters]	8.19 ± 4.57	2.18 ± 1.55	<0.001 ^a^
Axial length	24.00 ± 3.85	23.47 ± 1.08	0.715 ^b^

Results are shown as mean and standard deviation. ^a^ Paired *t*-test; ^b^ Wilcoxon signed-rank test. BCVA—best corrected visual acuity; logMAR—logarithm of the minimum angle of resolution.

**Table 2 children-10-01519-t002:** Refractive errors in the children.

Serial Number	Age	Sex	Amblyopic Eye			Fellow Eye		
			Sphere	Cylinder	Axis	Sphere	Cylinder	Axis
1	11	F	−6.00	3.62	170	1.75	3.12	5
2	11	M	8.87	−1.25	10	1.25	0.00	0
3	11	M	6.76	−1.62	180	2.00	−0.25	160
4	11	M	−17.50	−4.25	105	−3.12	−1.25	30
5	13	M	−12.00	−3.12	175	0.25	−0.62	175
6	14	M	5.25	2.62	70	1.37	0.25	105
7	15	M	7.62	−0.12	140	3.12	−0.25	80
8	17	M	6.62	−1.62	180	0.75	0.00	0
9	17	M	−2.87	−2.25	180	3.12	−0.87	5
10	18	M	6.00	−0.62	170	0.87	−0.37	15
11	10	M	9.87	−0.75	140	6.12	−0.25	15
12	14	M	4.00	0.00	0	2.50	−0.62	170
13	16	M	−13.62	−0.75	55	−0.62	−0.62	175
14	9	F	5.50	0.87	140	2.62	0.00	0

F—female, M—male.

**Table 3 children-10-01519-t003:** Comparison of central superficial vessel density, foveal avascular zone [FAZ] in the superficial capillary plexus [SCP], and central macular full retinal thickness [TCS] between amblyopic and fellow eyes.

	Amblyopic Eyes	Fellow Eyes	*p*-Value
VD Full [%]			0.328 ^b^
Mean ± SD	18.32 ± 1.22	18.90 ± 0.62	
Median [range]	18.80 [15.20 to 19.30]	19.00 [17.80 to 19.70]	
FAZ SCP [mm^2^]			0.808 ^a^
Mean ± SD	0.227 ± 0.07	0.232 ± 0.10	
Median [range]	0.21 [0.13 to 0.35]	0.22 [0.09 to 0.45]	
TCS [µm]			0.291 ^a^
Mean ± SD	261.50 ± 36.63	254.50 ± 23.14	
Median [range]	259.50 [197.00 to 347.00]	258.00 [215.00 to 288.00]	

VD—vessel density; SD—standard deviation; FAZ—foveal avascular cone; SCP—superficial capillary plexus; TCS—thickness central subfield. ^a^ Paired *t*-test; ^b^ Wilcoxon signed-rank test.

**Table 4 children-10-01519-t004:** Vessel density in superficial capillary plexus and macular thickness at different locations between amblyopia and fellow eyes.

Location	Amblyopic Eyes	Fellow Eyes	*p*-Value
Macular thickness [µm]			
IS	322.00 ± 38.42	313.07 ± 21.62	0.159 ^a^
OS	285.64 ± 37.27	274.00 ± 14.97	0.123 ^a^
IN	319.79 ± 31.22	318.14 ± 19.60	0.736 ^a^
ON	299.00 ± 31.42	294.64 ± 22.19	0.267 ^a^
II	319.86 ± 39.82	306.14 ± 36.69	0.140 ^b^
OI	279.00 ± 40.07	267.71 ± 21.87	0.133 ^a^
IT	313.71 ± 42.58	302.14 ± 22.77	0.140 ^b^
**OT**	**281.00 ± 34.23**	**258.29 ± 18.07**	**0.010 ^a^**
C	261.50 ± 36.63	254.50 ± 23.144	0.291 ^a^
VD in SCP [%]			
IS	17.99 ± 1.27	18.75 ± 0.92	0.169 ^b^
OS	18.89 ± 0.10	19.03 ± 0.99	0.404 ^a^
IN	17.67 ± 1.78	18.17 ± 1.44	0.432 ^b^
ON	19.52 ± 0.94	19.99 ± 0.77	0.154 ^a^
II	18.14 ± 1.45	18.57 ± 1.13	0.259 ^a^
OI	18.94 ± 0.70	19.06 ± 0.90	0.231 ^a^
IT	17.72 ± 2.48	18.55 ± 1.00	0.332 ^b^
OT	17.31 ± 2.53	18.35 ± 0.57	0.357 ^b^
C	10.44 ± 2.80	11.72 ± 2.96	0.084 ^a^

IS—inner superior; OS—outer superior; IN—inner nasal; OS—outer nasal; II—inner inferior; OI—outer inferior; IT—inner temporal; OT—outer temporal; C—central; VD—vessel density; SCP—superficial capillary plexus. Results are shown as mean and standard deviation. ^a^ Paired *t*-test; ^b^ Wilcoxon signed-rank test.

## Data Availability

Not applicable.

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
