# Peer review of "OCT-Angiography Findings in Children with Anisometropic Amblyopia"

_children, 2023, doi:10.3390/children10091519_

Round 1

Reviewer 1 Report

The authors did not adequately defend the rational for the investigation.  It is a stretch to believe that the retina would be affected in anisometropia.  

For the manuscript title, anisometropia is defined as having one amplyopic eye.  As such, the word unilateral in the title is not needed. 

TCS should be defined as an acronym in the abstract.

In the results section, the paragraph begins with 14 eyes of 14 children.  Both eyes were tested.  I assume that the authors mean that the 14 ambloyic eyes were subsequently studied.  This should be rewritten as it is not clear.  Is the non-amblyopic eye the control eye in the analysis?

Was a power calculation done so that 14 subjects is sufficient?  The final sentence on page 3 is duplicated on the top of page 4.

If a finding would have been identified, what would be impact on clinical practice?  Would it matter at all?

Author Response

Point 1: The authors did not adequately defend the rational for the investigation.  It is a stretch to believe that the retina would be affected in anisometropia.                           

Response 1: We added to the aim of our investigation reviewer recommendations. Till now, previous studies revealed different findings in amblyopic eyes on retinal thickness with no changes on retinal thickness to larger FAZ area with non- to significant changes of lower vascular density of SCP. The aim of our study was to contribute to this till now not clear enough established field of OCT-angiography findings in amblyopic eyes.  For this propose we included the eyes with severe anisometropia (4 D or more) for better evaluation of OCTA findings between anisometropic ambyopic and non-amlyopic eye.

Point 2: For the manuscript title, anisometropia is defined as having one amplyopic eye.  As such, the word unilateral in the title is not needed.                                          

Response 2: The comment for manuscript title is correct. As mentioned in Introduction part anisometropia is defined as a condition of asymmetric refraction between the two eyes. This condition is defined by a difference of 1 or more diopters in spherical equivalent. The term anisometropic amblyopia is reserved for amblyopia in eyes with severe refractive error compared to other eyes without other pathologic conditions. In our study only the eyes with severe refractive error to other eye were included in the study (4 D or more). So we changed the title following reviewer recommendation to “OCT-angiography findings in children with anisometropic amblyopia”.

Point 3: TCS should be defined as an acronym in the abstract.                                          

Responce 3: TCS is now defined in the abstract as central macular full retinal thickness.

Point 4: In the results section, the paragraph begins with 14 eyes of 14 children.  Both eyes were tested.  I assume that the authors mean that the 14 ambloyic eyes were subsequently studied.  This should be rewritten as it is not clear.  Is the non-amblyopic eye the control eye in the analysis?                                                                                                    Response 4: Yes, all 14 ambyopic eyes were subsequently studied. But we also tested the fellow eyes who serve us as a control group in our study. This part was rewritten for better understanding. It was also added in the Material and Methods for better understanding.

Point 5: Was a power calculation done so that 14 subjects is sufficient?  The final sentence on page 3 is duplicated on the top of page 4.                                                                                               

Response 5: We included in our study only children with severe anisometropia (4 D or more). Of course, the number is not sufficient, but we included only children with severe anisometropia with unilateral amblyopia to be sure to find out any changes in retina if they exist.

The final sentence on page 3 is not a duplicate, because the first is average refraction expressed in sphere and the other sentence is average refraction expressed in spherical equivalent, but duplicate are the numbers. So, we corrected the value in sphere.

Point 6: If a finding would have been identified, what would be impact on clinical practice?  Would it matter at all?                                                             Response 6: If our results revealed the changes in retinal structure on OCTA we can conclude that amblyopia in cases with anisometropia can be associated with structural changes of the retina, especially with vascular density in SCP. So we could conclude that morphological changes predispose or contribute to amblyopia. But in our study as in some others this hypothesis was not confirmed.

Reviewer 2 Report

Pekica et al. evaluated the features of OCTA in 14 children with unilateral anisometric amblyopia, regardless of the type of refractive disorder, with a statistic significant difference regarding the sphere and spheric equivalent between the amblyopic and the fellow eye.

The manuscript has some major limitations and needs some improvements, as pointed out below; 1. Limited sample size which has not been mentioned in the limitation section is an important major limitation of the study. 2. SCP and DCP measurements should be defined in a more detail manner in method section. 3. There are some typos and grammatical errors in the manuscript.
4. The figure is not clear enough.
5. The discussion section is not sufficiently substantial and the structure is not clear.

Moderate editing of English language required

Author Response

Point 1: Limited sample size which has not been mentioned in the limitation section is an important major limitation of the study.

Response 1: The number of patients is small and this is an important limitation factor. In the majority of the studies for OCTA findings in amblyopic eye the number of children was small. But it must be emphasized that the limited sample size is one of important limitation in our study. The selection of our patients was well defined regarding the amount of refractive error between amblyopic and non-amblyopic eye. Difference between amblyopic eye to the non-amblyopic eye for 4 or more diopters was important inclusion criteria. The other eye in all participants must be non-amblyopic with visual acuity at less 0,8 or more with correction.

Point 2: SCP and DCP measurements should be defined in a more detail manner in method section.

Response 2: In Method section the SCP and DCP were more precisely defined.

SCP was defined with an inner boundary of internal limiting membrane and an outer boundary at the inner plexiform layer. This measurement is performed automatically by a built-in AngioPlex software.  In this way, we were able to ensure objective data. The data from internal limiting membrane to inner plexiform layer/inner nuclear layer interface for SCP was independent from investigator. We did not use the data from DCP (from the inner plexiform layer/inner nuclear layer interface to the outer plexiform layer/outer nuclear layer interface) because they are not performed automatically and are not included into the AngioPlex software. In our study we want to use only these parameters that can be exactly measured without any influence.

Point 3: There are some typos and grammatical errors in the manuscript.

Response 3: The typos and grammatical errors in manuscript were renewed corrected.

Point 4: The figure is not clear enough.

Response 4: The data of the Figure 1 was more precisely presented to avoid obscurity.

Point 5: The discussion section is not sufficiently substantial and the structure is not clear.

Response 5: The discussion section was changed regarding the reviewer recommendations. It is now structured and clearly.

Point 6: Comments on the Quality of English Language.  Moderate editing of English language required

Response 6: The editing of English language was performed to correct errors in the paper.

Round 2

Reviewer 1 Report

Edits are acceptable 

Author Response

Thank you very much for your kind review.

I concluded that you agree with new revised manuscript and you have no recommendation anymore.

Reviewer 2 Report

My suggestion is that the authors continue to revise some grammatical problems in the manuscript. In addition, it is better to increase the sample size.

My suggestion is that the authors continue to revise some grammatical problems in the manuscript. In addition, it is better to increase the sample size.

Author Response

Thank you very much for your kind review.

We are aware that the most important limitation of our study was a small number of included children. As we already mentioned in the Discussion, we excluded all anizometropic eyes with refractive error less than 4 dioptres for better detection the difference of OCT-angiography changes between amblyopic and fellow eyes.

 In the future we want to perform another study with children with smaller anisometropia ranging from 1 to 2 diopters and higher number of included children. So, in this case we can also expect different results.

We included this additional information in new revised manuscript.